# Altruistic punishment does not increase with the severity of norm violations in the field

Loukas Balafoutas[1], Nikos Nikiforakis[2] & Bettina Rockenbach[3]

The degree of human cooperation among strangers is a major evolutionary puzzle. A prominent explanation is that cooperation is maintained because many individuals have a predisposition to punish those violating group-beneficial norms. A critical condition for cooperation to evolve in evolutionary models is that punishment increases with the severity of the violation. Here we present evidence from a field experiment with real-life interactions that, unlike in lab experiments, altruistic punishment does not increase with the severity of the violation, regardless of whether it is direct (confronting a violator) or indirect (withholding help). We also document growing concerns for counter-punishment as the severity of the violation increases, indicating that the marginal cost of direct punishment increases with the severity of violations. The evidence suggests that altruistic punishment may not provide appropriate incentives to deter large violations. Our findings thus offer a rationale for the emergence of formal institutions for promoting large-scale cooperation among strangers.

[1] Department of Public Economics, University of Innsbruck, Universitätsstraße 15, A-6020 Innsbruck, Austria. [2] Division of Social Science, New York University Abu Dhabi, P.O. Box 129188 Abu Dhabi, UAE. [3] Department of Economics, University of Cologne, Universitätsstraße 22a, D-50923 Köln, Germany. Correspondence and requests for materials should be addressed to B.R. (email: bettina.rockenbach@uni-koeln.de).

The degree of human cooperation among strangers in large societies is a major evolutionary puzzle[1,2]. One of the most prominent explanations is that cooperation is maintained because many individuals have a predisposition to punish those violating group-beneficial norms, even if doing so is costly[1–3]. A critical condition for cooperation to be sustained in evolutionary models is that, as in formal institutions charged with maintaining social order[4], altruistic punishment 'fits the crime', that is, it increases with the severity of the violation[5–12]. Although this condition appears to be satisfied in lab experiments[13–18], the role of altruistic punishment in maintaining cooperation in daily life cannot be established without field data[2,3]. Controlled experiments have recently been carried out in natural field settings documenting the predisposition to punish norm violators[19,20], but there is no evidence to date on whether altruistic punishment is responsive to the severity of norm violations.

Here we present findings from the first field experiment investigating whether punishment 'fits the crime' in real-life interactions. The experiment was conducted at the two largest train stations in Cologne, Germany, to ensure that interactions were most likely one-shot. A team of confederates violated the social norm of not littering in public areas under different experimental conditions which differed only in the severity of the norm violation and the means available for punishing. The 'small violation' features a confederate ('Violator') noticeably throwing an empty coffee cup on the train platform close to a passenger, while in the case of the 'large violation' the item thrown on the platform was a large paper bag, including among others the same empty coffee cup (Fig. 1). A confederate recorded the reaction of a passenger who was standing alone close to the violation and who observed the violation ('Observer') (for more details see 'Notes on the experimental procedure' in the Methods). The main variables of interest are whether the Observer punished the Violator, for example, by asking her to pick up the litter or reprimanding her for her action, as well as the intensity of the punishment in cases when it occurred. The intensity of punishment was classified by independent coders as being low, medium, or high based on the exact transcripts collected by our team of research assistants (see 'Intensity of direct punishment' in the Methods).

Punishment of norm violators can occur directly by means of confronting the violator as in the two treatments described above. Direct punishment is typically costly because it requires time and effort to enact, and the punisher bears the risk of retaliation[21–24], which is why such punishment has been termed 'altruistic'[13]. However, punishment can also occur indirectly by means of withholding help[20]. Although helping requires effort, indirect punishment can also be costly, for instance when there are norms for helping others or when those withholding help suffer a reputational cost[1]. To investigate the possibility that indirect punishment responds to the severity of the norm violation, we implemented for each violation severity two treatments. In 'Small Help' and 'Large Help' the Violator, a few seconds after

having thrown the litter, reached inside her bag and a pack of books accidentally fell out. This allowed the Observer to offer help to recover one or more books. Hence, in these two treatments, the Observer had the option to punish the Violator directly, punish indirectly by withholding help, punish both ways or not punish at all. If helping rates are smaller in the 'Large Help' treatment than in the 'Small Help', then we will have evidence that indirect punishment increases with the severity of the violation. The two treatments 'Small No Help' and 'Large No Help' are identical in the violation act, but lack the book-falling act.

Table 1 summarizes our experimental treatments. In each of the four treatments we gathered at least 100 observations. The sample size was chosen such that if the propensity to punish large violations relative to small violations in the field is similar to that observed in lab experiments[13,14], we would detect the difference in punishment rates at the 5% level of significance with a very high probability (see 'Power calculations and sample size' in the Methods).

Additionally, we administered two surveys in the same location, but with different people (Supplementary Methods and Supplementary Note 1). In the first survey—conducted parallel to the experiment—respondents were presented with one of the two violations (small or large), and asked how they would feel about it, how they would react to it (for example, punish, withhold help), as well as the reasons for their reactions ($N = 510$). In the second survey—administered after the experiment—respondents were presented with both violations and asked to make direct comparisons between the two[25] ($N = 324$). These surveys allow establishing that our treatment manipulation was successful and help us better understand individual motives in the experiment.

The experimental evidence suggests that a key condition in all models seeking to explain the evolution of cooperation by means of altruistic punishment is not met in the field: altruistic punishment does not increase with the severity of the norm violation. This finding holds both for direct punishment (confronting a violator) and indirect punishment (withholding help). We furthermore document growing concerns for counter-punishment as the severity of the violation increases, indicating that the marginal cost of direct punishment increases with the severity of the violation. Our findings suggest that altruistic punishment may not provide appropriate incentives to deter large violations.

## Results

**Treatment manipulation.** We use survey responses to evaluate whether passengers consider overall the large violation to be more severe than the small violation, whether the large violation triggers stronger negative emotions, and whether respondents believe the large violation should be more strongly reprimanded. In the first survey, when asked independently how bothered they would be by each violation, respondents were significantly more bothered by the large than the small violation (Mann–Whitney U, $z = -3.059$, $P = 0.002$, $N = 510$). In the second survey, when asked directly to compare the two violations, passengers were significantly more bothered by the large violation (Wilcoxon sign-rank, $z = 9.39$, $P < 0.001$, $N = 324$) and judged it as being significantly more severe (Wilcoxon sign-rank, $z = 6.621$, $P < 0.001$, Wilcoxon sign-rank, $N = 324$). This finding is important as negative emotions have been shown to be the proximate mechanism for altruistic punishment[13]. Finally, when asked whether small and large violators should be equally reprimanded or whether one of them should be more strongly reprimanded, respondents in the second survey stated on aggregate that large violators should be more strongly reprimanded (Wilcoxon

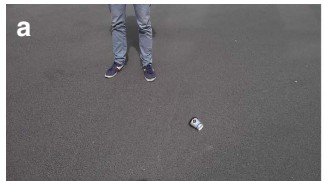
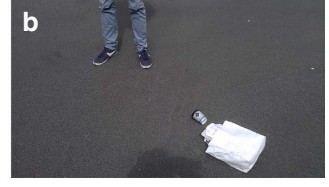

**Figure 1 | Pictures of the violations.** Panel (**a**) shows the small violation: littering by throwing a small coffee cup. Panel (**b**) shows the large violation: littering by throwing a lunch bag with several items, including a small coffee cup.

**Table 1 | Experimental treatments.**

|  | Small no help | Large no help | Small help | Large help |
|---|---|---|---|---|
| *Act 1*: Norm violation | Violator throws coffee cup | Violator throws paper bag with coffee cup | Violator throws coffee cup | Violator throws paper bag with coffee cup |
| *Act 2*: Needing help | – | – | Violator's books fall out of bag | Violator's books fall out of bag |
| *Dependent Variable* | Direct punishment | Direct punishment | Direct punishment & helping | Direct punishment & helping |
| N | 102 | 100 | 100 | 102 |

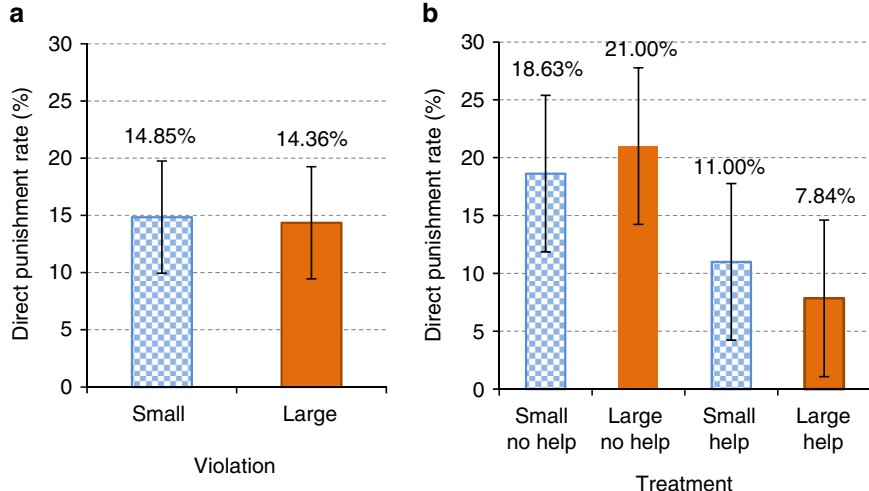

**Figure 2 | Direct punishment rates.** Panel (**a**) compares the rate of direct punishment across treatments with small and large violations. The difference in punishment rates for small and large violations is not statistically significant ($\chi^2(1) = 0.020$, $P = 0.89$, $N = 404$). Panel (**b**) presents the rate of direct punishment by treatment. The difference is statistically significant neither in the treatments without helping opportunities ($\chi^2(1) = 0.179$, $P = 0.67$, $N = 202$) nor the treatments with helping opportunities ($\chi^2(1) = 0.591$, $P = 0.44$, $N = 202$). Direct punishment rates are significantly lower in the treatments with helping opportunities than in those without ($\chi^2(1) = 8.753$, $P = 0.003$, $N = 404$). The bars indicate 95% confidence intervals.

sign-rank, $z = 5.06$, $P < 0.001$, $N = 324$). Hence, our treatment manipulation succeeded in creating two conditions that differ in the severity of the norm violation as perceived by the population under consideration, the extent of negative emotions they trigger, and the extent people believe the violators should be reprimanded. If punishment 'fits the crime' in the field, we expect to observe more punishment for the large violation than for the small violation.

**Direct punishment**. The rates of direct punishment across treatments can be seen in Fig. 2. Strikingly, despite the fact that passengers report significantly stronger negative emotions towards the large than the small violation and believe large violators should be more strongly reprimanded, direct punishment rates are very similar for the large and small violation, and never significantly different from each other. This pattern is true for the aggregate of the 'No Help' and the 'Help' treatments (14.85 versus 14.36%; $\chi^2(1) = 0.020$, $P = 0.89$, $N = 404$; see Fig. 2a) as well as for the treatments separately (No Help: 18.63 versus 21.00%; $\chi^2(1) = 0.179$, $P = 0.67$, $N = 202$; Help: 11.00 versus 7.84%; $\chi^2(1) = 0.591$, $P = 0.44$, $N = 202$; see Fig. 2b). Moreover, the intensity of direct punishment, that is, the way the norm violator is reprimanded, is not significantly different across violations ($\chi^2(2) = 1.024$, $P = 0.60$, $N = 59$; see Fig. 3 and 'Intensity of direct punishment' in the Methods). Thus, we provide clear evidence that in the field direct punishment does not increase with the severity of the norm violation.

**Indirect punishment**. We first note that direct punishment rates are lower in the treatments with helping opportunities (19.80% in

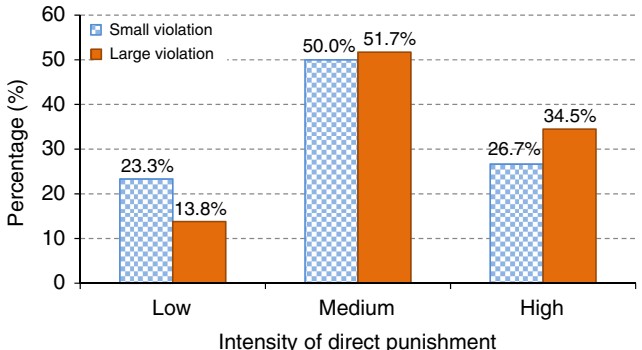

**Figure 3 | Intensity of direct punishment by type of violation.** The figure shows the incidences of low, medium and high intensity of direct punishment for small and large violations. The difference in the intensity of punishment for small and large violations is not statistically significant ($\chi^2(2) = 1.024$, $P = 0.60$, $N = 59$). The coding is explained in 'Power calculations and sample size' in the Methods.

'No Help' versus 9.41% in 'Help'; $\chi^2(1) = 8.753$, $P = 0.003$, $N = 404$), indicating that the possibility of indirect punishment by means of withholding help crowds out the willingness to punish violators directly, as in (20). In the absence of a violation, the helping rate in (20)—measured in the same location and under identical conditions—was 42.9%. This rate is significantly higher than that in 'Small Help' ($\chi^2(1) = 20.162$, $P < 0.001$, $N = 177$), and in 'Large Help' ($\chi^2(1) = 22.539$, $P < 0.001$, $N = 179$) indicating the use of indirect punishment in our experiment. Importantly,

helping rates for large and small violators do not differ (13.00% in 'Small Help' and 11.76% in 'Large Help'; $\chi^2(1) = 0.071$, $P = 0.790$, $N = 202$) suggesting that indirect punishment also does not 'fit the crime' in our experiment.

## Discussion

Lab evidence shows that the total amount individuals are willing to invest in punishing strangers increases with the severity of the violation[13–18]. The proximate mechanism for punishment is negative emotions triggered by the violation[13]. Since we observe a similar increase in magnitude in negative emotions in our experiment (see 'Negative emotions in response to norm violations' in the Methods), we should expect more punishment towards the large violators. Yet, we have seen that altruistic punishment does not increase with the severity of the violation in the field. Punishment rates do not differ, neither for direct nor for indirect punishment. Thus, there must be a fundamental difference between punishing in existing lab experiments and in our field experiment.

With regards to direct punishment, most respondents in our first survey who stated that they would not punish despite being bothered by the violation, stated that this was out of fear for being counter-punished by the violator (given by a total of 60% of respondents in that group). Counter-punishment raises the cost to punishers for enforcing cooperation[22]. Importantly, the fear of counter-punishment increases with the severity of the norm violation. As seen in Fig. 4, 53.7% of respondents said they would not punish small violations for fear of counter-punishment, compared with 67% of respondents in the case of large violations ($\chi^2(1) = 3.884$, $P = 0.049$, $N = 211$). Thus, unlike in most lab experiments where the marginal cost of punishment is either constant[13,16–18] or decreasing[14,15], in the field the (perceived) marginal cost of punishment appears to increase with the severity of the violation. This evidence may reflect a belief that more severe violations convey information about the social orientation of the violator, their general disregard for social attitudes and therefore the likelihood to engage in other types of antisocial behaviour such as counter-punishing. For models of cooperation, this finding implies an increasing evolutionary disadvantage for punishers[26], making direct punishment less likely to prevent large violations.

How can we explain the fact that indirect punishment is also unresponsive to the severity of violations in our experiment? Unlike direct punishment, it seems unlikely that indirect

punishment triggers fears of counter-punishment. Indeed, examining the open-ended responses to our first survey about why people would help violators or not, there was not a single case a respondent stated s/he would help (that is, not punish indirectly) out of fear of retaliation. The responses suggest respondents generally fall into one of two categories: those who withhold help in order to punish violators no matter how small the violation is, and those who would help both violators because 'that's the right thing to do'. To investigate this conjecture, in the second survey, respondents presented with the two violations were asked whether they would be more likely to help the small violator, the large violator, neither of the two, or both. Of the respondents who stated they would help at least one of the violators, 97.3% said they would be equally likely to help small and large violators. Of the respondents who said they would withhold help from at least one of the violators, 90.3% stated they would withhold help from both violators. Even if we limit our sample to respondents explicitly stating they would be more bothered by the large violation, 96.4% would treat the two violators equally: 74.6% said they would withhold help from both, and 21.8% said they would help both. This finding suggests that helping rates are not sensitive to the severity of the violation because, even though large violations are considered to be worse, small violations are sufficient to justify withholding help.

In summary, we present the first evidence to suggest that a key condition in all models seeking to explain the evolution of cooperation by means of altruistic punishment is not met in the field: altruistic punishment does not increase with the severity of the violation. This evidence indicates that altruistic punishment may not provide appropriate incentives for strangers to cooperate. This finding is the more remarkable as we studied a population characterized by strong norms of civic cooperation for which punishment has been shown to 'fit the crime' in the lab[27]. Our results thus provide a rationale for the evolution of mechanisms that reduce the threat of counter-punishment and, generally, provide appropriate incentives for cooperation such as pooled punishment[28–30] and formal institutions charged with maintaining social order among strangers. In addition, our findings indicate that counter-punishment should not be ignored in theoretical or experimental analyses. Although a few pioneering studies have explored the related concept of anti-social punishment[31–33], more theoretical work is needed in order to understand how the willingness to retaliate punishment may have evolved[34] and its implications for the evolution of cooperation.

## Methods

**Notes on the experimental procedure.** All aspects of the study, including ethical acceptability, were reviewed by the Vice-Rectorate for Research at the University of Innsbruck and permission was granted to conduct the experiment. The Deutsche Bahn gave written consent to running the experiment, which took place in May 2015 on various platforms in the two large (long-distance) train stations in Cologne, Germany. The data collection occurred between 9 am and 3 pm. The data were collected in five groups of two confederates each (one actor and one supervisor). All actors were female. The actors performed the violations by throwing an item on the platform so that a nearby observer saw it. The violators threw the item in an obviously intentional, but not provocative manner. Approximately 10–15 s later, in treatments 'Small Help' and 'Large Help', the actor 'accidentally' dropped some books in front of the selected observer (while trying to get something from the bottom of her shoulder bag).

Acts were performed only with single, standing observers to ensure there was no second-order public good problem with respect to punishing the violator and that helping was costly for individuals who had to bend to pick up the books. Observers were randomly assigned into treatments. Kruskal-Wallis tests using the observer's gender, height, age, and the confederates, suggest the randomization was successful (gender: $P = 0.95$; height: $P = 0.95$; age: $P = 0.14$; confederates: $P = 0.95$). The $P$-values are from two-tailed tests, like all $P$-values reported in the paper.

Supervisors were standing at a distance of about 3–5 metres away from the scene of the violation and were instructed to remain passive during the entire interaction, and avoid any interference. Supervisors recorded only acts in which the

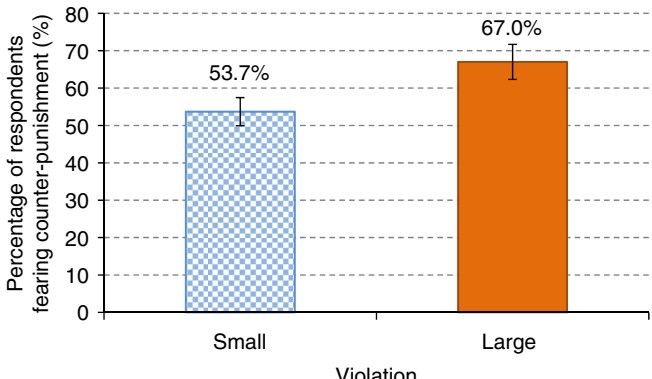

**Figure 4 | Proportion of respondents fearing counter-punishment.** The figure shows the proportion of respondents who stated fear of counter-punishment as the reason why they would not apply direct punishment for each violation (Survey 1, question 4). The proportion is significantly higher for large violations ($\chi^2(1) = 3.884$, $P = 0.049$, $N = 211$). The bars indicate 95% confidence intervals.

observer did not leave the scene before the scripted act was complete and no other passenger approached. Observing the violation and then leaving the scene was coded as no punishment. Supervisors also ensured the acts were performed according to the script, and that observers witnessed the behaviours described in Acts 1 and 2 (Table 1). Supervisors recorded whether the observer helped to pick up the books (in treatments Small Help and Large Help), whether he or she applied direct punishment against the violator (in all treatments), and the exact form that direct punishment took. Whenever an observer picked up at least one book, his or her action was recorded as help. Whenever an observer explicitly asked the violator to pick up the cup, or expressed disapproval of the norm violation his or her action was recorded as direct punishment.

Supervisors also recorded the following information: time of day when the observation was collected, an estimate of the observer's age, activity of observer while waiting (for example, eating, reading, just waiting), time to next train, nonverbal response of observer to violation, reaction to dropping the books. After the interaction was completed the team picked up any litter discretely and moved to a different platform. All supervisors and actors were blind to the research hypotheses.

The first survey was administered in the same location as the experiment and parallel to it, in May 2015. The second survey was administered also in the same location in June 2016.

**Intensity of direct punishment.** The supervisors were instructed to precisely record the form of direct punishment by observers (if any), immediately after punishment occurred. We use these records to classify the intensity (severity) of direct punishment. We recruited two German-speaking research assistants who were unrelated to our study and blind to the hypotheses. They were asked to independently classify each reported instance of punishment as being of either 'low', 'medium' or 'high' intensity. After producing independent ratings, the two assistants met and discussed cases of disagreement, in order to obtain a final coding. The Spearman rank correlation coefficient between their respective ratings was 0.52 ($P < 0.001$).

Figure 3 shows the incidences of low, medium and high intensity of direct punishment for small and large violations, using the final coding. The difference is not statistically significant ($\chi^2(2) = 1.024$, $P = 0.60$, $N = 59$). We note that our findings regarding the statistically indistinguishable intensity of direct punishment across treatments are robust when we consider each assistant's pre-agreement coding separately (Assistant 1: $\chi^2(2) = 1.136$, $P = 0.57$, $N = 59$; Assistant 2: $\chi^2(2) = 0.431$, $P = 0.81$, $N = 59$).

**Power calculations and sample size.** Convention prescribes that the sample size is such that a given treatment effect will be significant at the 5% level, 80% of the time (that is, the power of the test is 80%). Our study is the first to explore the relationship between the severity of norm violation and punishment in a natural field setting. We therefore have no prior on the potential size of the treatment effect (if any). To ensure that our statistical tests are sufficiently powered we used the following approach to calculate our sample size.

As a benchmark for the expected rate of direct punishment in treatment 'Small No Help' we use the direct punishment rate that we observed for women in treatment *BasePun* from Balafoutas et al.[20] where the violation and the experimental location were the same as in our study. That rate was 26.3%. In order to have an estimate of the expected rate of direct punishment with the large violation, we refer to observed differences in punishment rates between small and large violations in the two lab studies that introduced altruistic punishment in the literature, using a similar population. In Fehr and Gächter[14] individuals are willing to invest more than twice the amount they do for 'small violations' in order to punish 'large violations': taking into account the non-linear punishment technology, comparing small violations (that is, negative deviations of at most 8 tokens of the violator's contribution to the public good from others' average contributions) to large violations (that is, negative deviations of more than 8 points) reveals that the amount spent on punishment is on average about 2.25 times greater for large compared with small violations (Fehr and Gächter[14], p. 991). The same calculations using the data reported in Fehr and Gächter[13] lead to a factor of between 2.30 and 3.00. (As we show in section 'Negative emotions in response to norm violations' below, Fehr and Gächter[13] also use these ranges to distinguish between 'small violations' and 'large violations'. We also show that the large violation in our experiment triggers a similar increase in negative emotions.) We take a conservative approach and use the smallest of these factors, 2.25. This leads to an expected punishment rate of (26% × 2.25 = 58.5%) in treatment 'Large No Help'.

To minimize the likelihood of a Type-II error in case the treatment differences turned out to be smaller in the field (that is, failing to reject a false null hypothesis of no differences), we demanded a power of at least 99%. That is, if the rate of direct punishment in the field is as sensitive to the relative severity of the violation as in the lab, then we would be almost certain to detect it at the 5% level. If our statistical results fail to detect a significant difference, therefore, this could not be attributed to our tests being underpowered. Given the aforementioned punishment rates, the sample size needed to identify significant treatment differences at the 5% level with a power of 0.99 is estimated to be at least 160 observations in total, or $N = 80$ per violation. We aimed to collect approximately 100 observations per violation giving us a power of 99.8% given the assumptions above.

As it turned out, in our field experiment, we observe 18.63% of Observers directly punishing a norm violator in 'Small No Help'. The difference between the rate of 26.3% (15 out of 57 women) and 18.6% (19 out of 102) is not statistically significant ($\chi^2(1) = 1.286$, $P = 0.257$). If we had used the direct punishment rate of 18.6% for our power analysis, all else equal, we would expect a punishment rate of 41.85% ($= 18.6\% \times 2.25$) in 'Large No Help'. With a sample size of 100 in each treatment, this would imply a power of 96%.

**Negative emotions in response to norm violations.** Fehr and Gächter[13] argue that the punishment of strangers is likely to be propelled by the negative emotions triggered by the violation of social norms: "Free riding may cause strong negative emotions among the cooperators and these emotions, in turn, may trigger their willingness to punish the free riders. If this conjecture is correct, we should observe particular emotional patterns in response to free riding" (p. 139). In this section we explore whether the large violation in our experiment triggered stronger negative emotions than the small violation, and how any difference compares to that reported in Fehr and Gächter[13]. The study of Fehr and Gächter may be the best lab study to compare our findings with as they also investigate one-shot interactions between perfect strangers in a similar population, and the punishment of free riders benefits only other individuals in the future.

In Survey 1, respondents were asked to state how much they would be bothered by either the large or the small violation (question 2). Assigning numbers to each category ($0 = $ Not at all, $1 = $ Yes—a little, $2 = $ Yes—quite a lot, $3 = $ Yes—a lot), the weighted average is 2.194 for the large violation, and 1.964 for the small violation. This amounts to an 11.7% increase in negative emotions triggered by the large violation, which is statistically significant (Mann–Whitney U, $z = -3.059$, $P = 0.002$, $N = 510$).

This difference is very similar in magnitude to that reported in Fehr and Gächter[13]. Subjects in that study had to indicate how annoyed they would be by a group member in a public good game who contributed either 3 (small violation) or 14 (large violation) units less than his/her peers on average, using a seven-point scale ($1 = $ 'not at all' to $7 = $ 'very much'). (Notice that these numbers are consistent with our definitions of small and large violations in the previous section.) Fehr and Gächter[13] do not report the average response in their paper but they do write that: "It turns out that a free rider triggered much anger among the other subjects if these subjects contributed a lot relative to the free rider (scenario 1). Forty-seven per cent of the subjects indicated an anger level of 6 or 7 and another 37% indicated an anger level of 5. If the deviation of the free rider's contribution from the other members' contribution was relatively small (scenario 2), the anger level was significantly lower (Wilcoxon signed rank test, $z = 9.636$, $P < 0.0005$) but still considerable. In this case (scenario 2), 17.4% of the subjects indicated an anger level of 6 or 7 and 80.5% indicated a level of 4 or 5 in scenario 2".

Using the data from Fehr and Gächter[13], we find that for large violations (scenario 1) and for small violations (scenario 2) the average responses were 5.49 and 4.77 respectively. This amounts to a 15% increase, which is similar to the increase in negative emotions in our experiment (11.7%). Thus, a similar increase in negative emotions to that observed in our experiment led to a threefold increase in punishment in Fehr and Gächter[13].

**Data availability.** All relevant data are available from the authors.

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

## Acknowledgements

We thank the research assistants Jarid Zimmermann, Jenna Strzykala, Karen Heuermann, Marcin Waligora, Sebastian Schneiders, and Anne Schielke and the actors Christina Woike, Millie Vikanis, Mirjam Piesker-Muhl, Silvana Lepsa, and Verena Volland for their excellent support. We thank Simon Gächter for sharing the data from Fehr and Gächter (2002)[13]. We also thank Ernst Fehr, Manfred Milinski, David Rand, Matthias Sutter, and participants at the University of Cologne seminar, New York University Abu Dhabi seminar, the 1st NYU Global Network Workshop in Experimental Social Sciences, and the 85th Annual Meetings of the Southern Economic Association in New Orleans for useful comments.

## Author contributions

L.B., N.N. and B.R. designed the experiment, prepared the surveys, and wrote the manuscript, L.B. compiled the dataset and carried out the statistical analysis, B.R. trained and supervised the team of research assistants.

## Additional information

**Competing financial interests:** The authors declare no competing financial interests.

**Publisher's note**: 

