## [Peer Review File · Nature Communications]

Reviewers' comments:

Reviewer #1 (Remarks to the Author):

Title: Altruistic punishment does not "fit the crime" in the field

Summary: Models of altruistic punishment stabilizing cooperation depend on punishment severity increasing as violation severity increases. In a field study, participants were provided an opportunity to punish a norm violation (littering) that was either a small or large violation. In addition, participants in two conditions were given the opportunity to withhold help as a form of indirect punishment. Amounts of direct and indirect punishment did not differ between the small and large violations. Survey data indicates this is due to perceived increased costs of punishing for large violations.

This study was well written and I thought the set up was clever. The question of whether the marginal cost of punishment increases in a field setting is also interesting. However, I do have some major concerns that need to be addressed before I recommend publication.

1. The importance of the study rests on a null finding-that punishment does not differ between small and large violations. However, despite the author's assurances that the violations were different, I am not entirely convinced that the difference was sufficiently large enough to elicit different amounts of punishment. If it is the case that the null is due to the lack of a difference between the violations, then the finding is unimportant. The authors manipulate severity of littering by having the confederate discard a coffee cup (small violation) or a coffee cup and paper bag (large violation). Based on survey data showing that participants are more bothered by the coffee cup and paper bag, the authors conclude that they successfully manipulated severity with this manipulation. However, there are two problems with this. First, "bothered" is not the same as severity of the violation. A person can be bothered by the violation but still recognize the violation is not significantly more severe. Second, evidence from criminological studies show that judgments of severity and punishment are logarithmic (Wolfgang et al., 1985), suggesting the severity manipulation may not have been large enough to elicit differential rates of punishment. Consider that wrongness and punishment judgments are made for acts ranging from littering to mass murder - the difference between littering one and two pieces of trash is inconsequential in light of this variation and might not produce greater punishment.
2. Were the supervisors and confederates blind to the hypotheses? Were there any differences across the 5 groups of confederates/actors in rates of punishment for the different treatments that would suggest bias?
3. I understand the observers were randomly assigned to treatment groups. I would like the authors to provide data showing that the randomization worked (4e.g. make sure age, sex, race of observer and the confederate assigned to them was distributed equally across the groups).
4. I would have liked independent ratings for the severity of the punishment scale they used. For instance, non-verbal expressions of disapproval were rated "low intensity" along with watching the observer pick up the trash him or herself. I would think the latter would be moderate or even a high level of punishment by inducing shame or guilt.

5. Indirect punishment: The authors used a measure of (lack of) helping as a measure of indirect punishment. It is not clear from the design though if the low rates of punishment in both groups is due to a lack of punishment or simply a low base-rate of helping strangers. A control "no violation" help group, in which a confederate drops books but does not litter prior to this, would help to distinguish the two cases. If help does not differ between the no, small, and large violation group, this suggests the lack of help is not an imposition of indirect punishment.

6. An interesting finding from the article is that people expect greater costs of counter-punishment from more severe violations. People in fact are more likely to indicate tension as a reason for non-punishment for the large violation than the small violation. The authors state that someone who commits a large violation may be perceived as more "antisocial" and as having a "general disregard for social attitudes". If these are people who should be avoided, then why do the authors observe equal levels of helping in both the large and small violator groups? Also, it is not clear if "tension" measures counter-punishment or that the other answer categories do not measure counter-punishment as well. For example, the justification that other people do not reproach others could already take into account that other people do not reproach in fear of retaliation. Or not being nice could suggest a reputational concern, which would be a cost of counter-punishment.

7. Can the authors provide more detail about their methodology and specifically, how much time they let lapse before deciding to record an observer's behavior or lack thereof? For instance, they claim to have only recorded "those acts in which the observer did not leave the scene" but leaving the scene be recorded as no punishment? Should there be a "no punishment" level? How would the results change if this were included? I can also imagine that many people, in order to avoid having to punish, pretend to not see violations. Can the authors comment on this too? I guess a coffee cup falling to the ground is easier to pretend not to see... so is it possible that observers pretended not to see the coffee cup fall and that there is actually less punishment of the small (coffee cup) violation that is being missed?

8. How close were the supervisors standing to the observers and actors? I imagine it is close enough that they can record accurately what is happening... on the other hand doesn't this introduce second order public good problems that the authors were trying to avoid? What are the implications of this?

References

Wolfgang, M.E., Figlio, R.M., Tracy, P.E., & Singer, S.I. (1985). *The National Survey of Crime Severity*. Washington, DC: U.S. Department of Justice, Bureau of Justice Statistics.

Reviewer #2 (Remarks to the Author):

The authors present an excellent field study on altruistic punishment behavior. They show very convincingly that altruistic punishment does not increase with the intensity of the social norm violation (as far as I am aware for the first time in a field experiment). On the contrary they report a tendency that more severe violations are less likely to be punished

directly, due to the fear of counter punishment. The presented study is methodically and statistically extremely well conducted. The findings are convincingly presented and discussed. The study is well set in the existing literature (except that they do not mention Fehl et. al. 2012 in the context of counter punishment).

I have only very short comments/questions which need clarification before publication:

1. In the supplementary information on page 1 (4.) the authors write ", whether he or she applied direct..." implying that that males and females were tested, however on page 7 , last paragraph it seems that they solely tested females. Also in the experiment description it is not fully clear to me whether the violator was always female or not. This should be clarified. (In comparison they have tested both sexes in the questionnaire.)

2. Also in the supplementary information there is a mistake on page 2. The percentages of the answers to question 3. and question 4. do not add up to 100% but to 103% and 97,48% respectively.

I strongly advise to publish the manuscript in nature communications.

Response to comments by Reviewer 1

We would like to thank you for the careful reading of our manuscript and your detailed comments. We have collected additional data and revised our paper to address all of your concerns. Your comments have helped to improve our paper. We are grateful for your advice.

Below, we copy your comments and proceed to address each of them in sequence.

This study was well written and I thought the set up was clever. The question of whether the marginal cost of punishment increases in a field setting is also interesting. However, I do have some major concerns that need to be addressed before I recommend publication.

We are very pleased you liked our research question and experimental set up, as well as how the paper was written up. We found all your concerns valid and adjusted the manuscript to address them. In some instances this required some minor clarifications on our part, while in others it required the collection of additional survey data.

We hope you will find the revised version satisfactorily addresses all of your concerns.

1. The importance of the study rests on a null finding - that punishment does not differ between small and large violations. However, despite the author's assurances that the violations were different, I am not entirely convinced that the difference was sufficiently large enough to elicit different amounts of punishment. If it is the case that the null is due to the lack of a difference between the violations, then the finding is unimportant. The authors manipulate severity of littering by having the confederate discard a coffee cup (small violation) or a coffee cup and paper bag (large violation). Based on survey data showing that participants are more bothered by the coffee cup and paper bag, the authors conclude that they successfully manipulated severity with this manipulation. However, there are two problems with this.

First, "bothered" is not the same as severity of the violation. A person can be bothered by the violation but still recognize the violation is not significantly more severe.

This is a valid point. To address it we designed and administered a new survey to 324 new passengers. Following Wolfgang et al. (2015), respondents were asked to compare directly the two violations. The evidence from the new survey illustrates that our treatment manipulation was indeed successful in generating two norms that our sample perceives to differ. In particular, passengers judged the large violation as being significantly more severe than the small violation ($p < 0.01$, Wilcoxon sign-rank, $N = 324$), were more bothered by the large violation ($p < 0.01$, Wilcoxon sign-rank, $N = 324$), and believed that large violators should be more strongly reprimanded ($p < 0.01$, Wilcoxon sign-rank, $N = 324$). We present these findings in the Results subsection called "Treatment manipulation". We hope this evidence and the associated discussion address your concern about the perceived severity of the two violations.

Second, evidence from criminological studies show that judgments of severity and punishment are logarithmic (Wolfgang et al., 1985), suggesting the severity manipulation may not have been large enough to elicit differential rates of punishment. Consider that wrongness and punishment judgments are made for acts ranging from littering to mass murder - the difference between littering one and two pieces of trash is inconsequential in light of this variation and might not produce greater punishment.

Thank you for the reference. It was very useful when thinking about your comment and designing the new survey. Accordingly, we have included it in our paper.

As mentioned above, we conducted a new survey to address this comment. Although indeed many respondents judged the two violations as being equally severe, a substantial fraction viewed the large violation as more severe. A Wilcoxon sign-rank test thus clearly rejects the hypothesis that the severity of the violations is perceived to be the same by our sample on aggregate ($p < 0.01$, two tailed, $N = 324$). More importantly, and in response to this second point you raise, respondents stated that large violators should be more strongly reprimanded ($p < 0.01$, Wilcoxon sign-rank, $N = 324$).

2. Were the supervisors and confederates blind to the hypotheses? Were there any differences across the 5 groups of confederates/actors in rates of punishment for the different treatments that would suggest bias?

The research associates were indeed blind to the research hypotheses. The table below presents punishment rates for each of the five teams. As can be seen, the punishment rates are similar across groups (mean = 14.60). The difference is not statistically significant (χ^2 , two-tailed, $p = 0.45$). We have added a sentence in the Methods section.

Team	mean (punishment rate)
1	.1625
2	.1482
3	.1899
4	.0864
5	.1446

3. I understand the observers were randomly assigned to treatment groups. I would like the authors to provide data showing that the randomization worked (4 e.g. make sure age, sex, race of observer and the confederate assigned to them was distributed equally across the groups).

This is an excellent point. Randomization seems to have worked perfectly. Below, you can see the means for the variables you mentioned (except 'race' which we didn't record given the ethnic homogeneity of our sample). Kruskal-Wallis tests for each variable indicate that, across treatments, we do not observe significant differences for any of the variables. We have noted this in the new Methods section.

treatment	mean(age)	mean(gender)	mean(height)	mean(team)
Large No Help	42.32	.51	1.75	3.00
Large Help	38.45	.51	1.73	3.01
Small No Help	38.94	.56	1.71	3.08
Small Help	37.35	.52	1.74	3.00
k-wallis	0.14	0.95	0.95	0.99

4. I would have liked independent ratings for the severity of the punishment scale they used. For instance, non-verbal expressions of disapproval were rated "low intensity" along with watching the observer pick up the trash him or herself. I would think the latter would be moderate or even a high level of punishment by inducing shame or guilt.

This is a fair point. To address it, we recruited two German-speaking research assistants who were unrelated to our study and blind to the hypotheses. They were asked to independently classify each instance of punishment as being of either “low”, “medium” or “high” intensity. The correlation between their respective ratings (Spearman’s $\rho=0.52$, $p<0.01$) is statistically significant. Following standard procedures in economics (see e.g., Cooper and Kagel, *American Economic Review*, 2005), after producing independent ratings, the two assistants met and discussed cases of disagreement, arriving at the final rating that we now use in our paper. Using the new rating(s) of the coders, we (again) find no significant difference in the intensity of punishment and the severity of the violation. In the SI, we also report our findings when using separately the pre-reconciliation ratings of the two assistants. Again, we find no significant difference in the intensity of punishment and the severity of the violation.

5. Indirect punishment: The authors used a measure of (lack of) helping as a measure of indirect punishment. It is not clear from the design though if the low rates of punishment in both groups is due to a lack of punishment or simply a low base-rate of helping strangers. A control "no violation" help group, in which a confederate drops books but does not litter prior to this, would help to distinguish the two cases. If help does not differ between the no, small, and large violation group, this suggests the lack of help is not an imposition of indirect punishment.

This is another valid point. It is true that a treatment in which an actor who has not violated a norm needs help is necessary to establish the existence of indirect punishment. In the revised version, we discuss data from such a treatment. The data was collected using the same research assistants, following identical experimental procedures, in the same locations, and drawing participants from the same general population for Balafoutas et al. (2014, PNAS). Importantly, this data – as the other experimental data reported in our paper – were collected prior to the events that occurred outside the main train station in Cologne, on New Year’s Eve 2016 (see e.g., <http://www.bbc.com/news/world-europe-35231046>). Since we cannot exclude the possibility that these events changed peoples’ attitudes towards (and interactions with) strangers, we feel that this data provide a better benchmark than newly collected data would.

In summary, the evidence provides support for the use of indirect punishment in our recent experiment. Using the subset of 77 observations from our previous study in which *female* research associates were in need of help (treatment BaseHelp, Balafoutas et al., 2014), we find that the mean helping rate was 42.9%. This helping rate is significantly different from the 13% helping

rate in *Small Help* ($N=177$, $p<0.0001$, two-tailed, χ^2) and the 11.8% helping rate in *Large Help* ($N=179$, $p<0.0001$, two-tailed, χ^2). We include these comparisons in the sub-section called “Indirect punishment” at the end of the Results section. We hope this addresses your concern.

6. An interesting finding from the article is that people expect greater costs of counter-punishment from more severe violations. People in fact are more likely to indicate tension as a reason for non-punishment for the large violation than the small violation. The authors state that someone who commits a large violation may be perceived as more "antisocial" and as having a "general disregard for social attitudes". If these are people who should be avoided, then why do the authors observe equal levels of helping in both the large and small violator groups?

This is an excellent observation. We apologize that we were not clearer in the original submission. Examining the open-ended responses to our first survey about why people would help violators, there was not a single case a passenger stated s/he would help (i.e., not punish indirectly) out of fear of retaliation. Therefore, the reason helping rates did not differ across violations could not be the fear of counter-punishment. The responses suggest that passengers fell into one of two categories: those that withhold help in order to punish violators no matter how small the violation is, and those that would help both violators because “that’s the right thing to do”. This could explain the similar, positive helping rates in the two treatments.

To investigate this conjecture further, in the second survey, respondents presented with the two violations were asked whether they would be more likely to help the small violator, the large violator, neither of the two or both. Of the respondents who stated they would help, 97.3% said they would be equally likely to help small and large violators. More importantly, of the respondents who said they would *not* help, 90.3% stated they would withhold help from both violators. Even if we limit our sample to respondents explicitly stating they would be more bothered by the large violation, 96.4% said they would treat the two violators equally: 74.6% said they would withhold help from both, and 21.8% said they would help both. This suggests helping rates are not sensitive to the severity of the violation because, even though large violations are considered to be worse, small violations are sufficient to justify withholding help (among those individuals who withhold help as a means of indirect punishment).

We have explained this in the Discussion section of our paper.

Also, it is not clear if "tension" measures counter-punishment or that the other answer categories do not measure counter-punishment as well. For example, the justification that other people do not reproach others could already take into account that other people do not reproach in fear of retaliation. Or not being nice could suggest a reputational concern, which would be a cost of counter-punishment.

This is a very insightful comment. Thank you. We have to apologize because we seem to have done a poor job translating the German text on this question.

The response to the question why they would not punish that we interpret as “counter-punishment”, in German, was: “*Weil das zu Streit führen könnte*” This translates to: “*Because*

this could lead to a dispute". This is the closest analogue to counter-punishment in daily life we could think of.

You are right that the second response ("because no one reproaches [violators]") could reflect a fear of counter-punishment. However, interpreting this response as relating to fear of counter-punishment would *strengthen* our results. In particular, the fear of counter-punishment from large (small) violators combining the two responses would be 72.1% (56.5%). Considering the first response alone, the fear of counter-punishment from large (small) violators is 67.3% (53.7%). That is, the difference increases further. We decided to take the conservative approach and interpret as "fear of counter-punishment" only the instances in which respondents gave the first answer, as we originally intended.

Finally, we note that the translation of the last sentence was also inaccurate. The German text was: "*Weil man Anderen nicht vorschreiben sollte, wie sie sich verhalten*". This translates to: "*Because no one should tell others how to behave*". We are now providing the corrected translations in the Supplementary Information of our paper.

7. Can the authors provide more detail about their methodology and specifically, how much time they let lapse before deciding to record an observer's behavior or lack thereof? For instance, they claim to have only recorded "those acts in which the observer did not leave the scene" but leaving the scene be recorded as no punishment? Should there be a "no punishment" level? How would the results change if this were included?

It appears that a sentence we had in the SI was not sufficiently clear, leading to the impression that instances in which the observer left the scene were discarded from the analysis. In particular, previously in the SI we wrote: "The supervisor recorded only those acts in which the observer did not leave the scene and no other passenger approached." Instead, what we should have written is: "*The supervisor recorded only those acts in which the observer did not leave the scene before the scripted act was complete and no other passenger approached.*" The underlined text has now been added in the Methods section.

What this means is that we only excluded observations in which, for instance, the observer left before the actress dropped her books. *Any* instance in which, according to the supervisor, the observer noticed the littering *was* used in the analysis. Observing the violation and then leaving the scene was coded as no punishment. This is now explicitly stated in the Methods section.

I can also imagine that many people, in order to avoid having to punish, pretend to not see violations. Can the authors comment on this too? I guess a coffee cup falling to the ground is easier to pretend not to see... so is it possible that observers pretended not to see the coffee cup fall and that there is actually less punishment of the small (coffee cup) violation that is being missed?

The supervisors were instructed to monitor carefully the observers, especially to establish that they witnessed the violation and the perpetrator. Observers were unaware of this so, whereas they could pretend to the actor that they did not see the violation, it would be much harder to fool the supervisors who tried to follow their eye movements. For our analysis, we use only observations

in which supervisors positively confirmed that the observer witnessed the violation. This was almost always the case given the nature of the violations.

Note that, as you mention, pretending would be more convincing for the small violation, as it is difficult to pretend not to see someone littering a bag. This however only strengthens our results as it implies that punishment rates could be higher for the small violations, all else equal.

8. How close were the supervisors standing to the observers and actors? I imagine it is close enough that they can record accurately what is happening... on the other hand doesn't this introduce second order public good problems that the authors were trying to avoid? What are the implications of this?

The supervisors were instructed to stand at a distance of about 3-5 meters away from the scene of the violation (we have now added this information to the Methods section). This means that they were always much farther away from the violation than the Observer, since by design the violation occurred close to the Observer. Moreover, the supervisors were instructed to remain passive during the entire interaction, and avoid any interference. Although we cannot rule out that the presence of supervisors may have an effect on the behavior of observers, it is crucial to note that any such effect would be held constant across treatments since the script and setting were always the same. This implies that the presence of supervisors has no bearing on our main findings, namely that direct and indirect punishment rates do not differ across violations.

Response to comments by Reviewer 2

We would like to thank you for the careful reading of our manuscript and for recommending the publication of our manuscript in Nature Communications.

Below, we copy your comments and proceed to address each of them in sequence.

The authors present an excellent field study on altruistic punishment behavior. They show very convincingly that altruistic punishment does not increase with the intensity of the social norm violation (as far as I am aware for the first time in a field experiment). On the contrary they report a tendency that more severe violations are less likely to be punished directly, due to the fear of counter punishment. The presented study is methodically and statistically extremely well conducted. The findings are convincingly presented and discussed. The study is well set in the existing literature (except that they do not mention Fehl et. al. 2012 in the context of counter punishment).

Thank you for your kind remarks. We are delighted to learn that you liked our study.

Thank you also for bringing to our attention the study by Fehl et al. (2012). We were not aware of it. We have added it in the paper.

I have only very short comments/questions which need clarification before publication:

1. In the supplementary information on page 1 (4.) the authors write ", whether he or she applied direct..." implying that that males and females were tested, however on page 7, last paragraph it seems that they solely tested females. Also in the experiment description it is not fully clear to me whether the violator was always female or not. This should be clarified. (In comparison they have tested both sexes in the questionnaire.)

You understood correctly that we only use female actors (i.e., violators) in our current study for the reasons explained in the paper. Observers were indeed from both genders. The passage you refer to at the bottom of page 9 in SI refers explicitly to "women" because in our *previous* study we used both female and male actors. For the power calculations for this study, for maximum comparability, we used only the observations from the female actors. We added the explicit reference to "women" at the bottom of page 7 to make this point salient. What might indeed have been confusing is that we added the same reference to "women" for the data from our current experiment. This was done to highlight that we are comparing "apples to apples". But we understand from your comment that this could be more confusing than helpful, so we decided to remove the second reference to "women". In particular, the passage has been adjusted as follows (the word stricken-through has been deleted):

"As a benchmark for the expected rates of direct punishment in treatment *Small No Help* we use the direct punishment rate that we observed for women in treatment *BasePun* from (20), where the violation and the experimental location were the same as in our study. That rate was 26.3%.

...

As it turned out, in our field experiment, we observe on average 18.63% directly punishing a norm violator in *Small No Help*. The difference between the 26.3% (15 out of 57 women) is not significantly different to the 18.6% (19 out of 102 women; $p=0.31$, two-tailed Fisher's exact).”

2. Also in the supplementary information there is a mistake on page 2. The percentages of the answers to question 3. and question 4. do not add up to 100% but to 103% and 97,48% respectively.

Thank you for pointing this error to us. We have corrected it. This was a typo and it does not affect our analysis or results.

REVIEWERS' COMMENTS:

Reviewer #1 (Remarks to the Author):

The authors did an excellent job addressing all of my concerns. If the authors wish to include the initial review and their responses in the supplementary materials I give my permission.

Congrats!

- Coren L Apicella (with input from Kristopher Smith).